# The Differences in the Diagnostic Profile in Children with Vasovagal Syncope between the Result of Head-Up Tilt Table Test

**DOI:** 10.3390/ijerph17124524

**Published:** 2020-06-23

**Authors:** Ewelina Kolarczyk, Lesław Szydłowski, Agnieszka Skierska, Grażyna Markiewicz-Łoskot

**Affiliations:** 1Department of Propaedeutics of Nursing, Faculty of Health Sciences in Katowice, Medical University of Silesia, 40-752 Katowice, Poland; 2Department of Pediatric Cardiology, Faculty of Medical Sciences in Katowice, Medical University in Silesia, 40-752 Katowice, Poland; szydlowskil@interia.pl (L.S.); aga.skierska@wp.pl (A.S.); 3Department of Nursing and Social Medical Problems, Faculty of Health Sciences in Katowice, Medical University of Silesia, 40-752 Katowice, Poland; mic54@o2.pl

**Keywords:** vasovagal syncope, children, tilt test, syncope, history taking, prodromal symptoms

## Abstract

(1) Background: The features characterizing vasovagal syncope (VVS) are an important factor in the correct evaluation of diagnostic risk stratification in children and adolescents. The aim of the study was to determine the value of identifying the clinical characteristics in children with VVS. (2) Methods: We made a retrospective analysis of the medical records of 109 children with diagnosed VVS. We investigated the specific characteristics of syncope in children with VVS including the positive VVS (+) and negative VVS (−) result of the Head-Up Tilt Table Test (HUTT). (3) Results: We did not observe significant differences in the prodromal symptoms of VVS with HUTT response. In addition to typical prodromal symptoms, no difference in statistically reported palpitations (35/109 or 32.1%) and chest discomfort (27/109 or 27.7%) were recorded. Fear–pain–stress emotions as circumstances of syncope were more often reported by children with a negative HUTT (*p* = 0.02). Cramps–contractures (*p* = 0.016) and speech disorders (*p* = 0.038) were significantly higher in the group with negative HUTT. (4) Conclusions: There is a close relationship in the diagnostic profile between the negative and positive results of head-up tilt table test in children with vasovagal syncope.

## 1. Introduction

Syncope is defined as a spontaneous and transient loss of consciousness (TLoc) due to a generalized decrease in cerebral perfusion. It is characterized by a rapid onset, short duration, and self-contained recovery that does not require intervention [1]. Syncope is a common clinical symptom in children and adolescents, which occurs with a frequency of 125 in 10,000 cases with a predominance of females [2,3]. It is estimated that approximately 40% of healthy teenagers have experienced at least one episode of syncope [4,5,6]. The most common loss of consciousness in young people include vasovagal syncope (VVS), known as reflexive, neurogenic, or neurocardiogenic. 

It is believed that in approximately 10% of children, determining the cause of syncope is unsuccessful [3,7,8]. The careful history taking of the characterizing clinical features of VVS, allow for the cause of syncope in the initial evaluation to be defined [1,3]. When a neurocardiogenic cause of syncope is suspected and heart diseases are excluded, a head-up tilt table test (HUTT) is used, due to which it is possible to confirm the diagnosis of VVS [9,10].

Syncope, especially of unknown etiology that can potentially be caused by organic diseases of the cardiovascular or nervous system, causes significant fear, both in patients and their families [11]. Apart from the reduction in quality of life of the patient, it may also be the cause of other injuries. Syncope may be the first symptom of ongoing cardiovascular and neurological disease. Verified knowledge of the causes and nature of syncope allows for correct diagnosis, avoiding the execution of expensive and highly specialized diagnostic tests [12,13]. The features characterizing VVS are an important factor in the correct evaluation of diagnostic risk stratification in children and adolescents [1]. Therefore, the aim of the present study was to determine the value of identifying the clinical characteristics in children with VVS.

## 2. Materials and Methods 

The data for the present study were collected in a retrospective analysis of the medical records of children hospitalized at the Children’s Cardiology Clinic at the Upper Silesian Children’s Health Centre in Katowice, Poland. We compared the characteristics of a group of children and adolescents with VVS with positive results of the head-up tilt table test, VVS (+), with those with negative results in the HUTT, VVS (−). 

### 2.1. The Studied Subjects

All patients affected by transient loss of consciousness were based on the initial examination standard according to the European Society of Cardiology guidelines [13]. All patients underwent an initial evaluation consisting of history taking, physical examination, orthostatic blood pressure, and standard 12 lead electrocardiogram. All patients were diagnosed to have VVS as a probable or certain cause of loss of consciousness. All children were referred for an exercise test, 24-h Holter monitoring, laboratory test, x-ray examination and/or computed tomography (CT), ultrasound (ECHO), and HUTT for a definite diagnosis or confirmation to the VVS. The cardiogenic, neurological, or psychological etiology of syncope was excluded on the basis of performed additional examinations in cases such etiology was suspected. 

In order to participate in this study, patients had to fulfill all the following inclusion criteria: a history of typical VVS with correct physical examination (with a special emphasis on cardiovascular diseases); a negative family history of sudden cardiac death episodes in relatives of a child before 30 years of age (sudden cardiac death of infants, drowning, car accidents in unexplained circumstances); a structurally normal heart, good physical activity tolerance in the exercise test according to the Bruce protocol; no cardiac arrhythmias in the resting electrocardiographic recording ECG and Holter test; normal laboratory results that exclude the presence of inflammation and ionic disorders (i.e., potassium, calcium, magnesium), and exclusion of a neurological or psychological cause of syncope.

### 2.2. Head-up Tilt Table Test 

The HUTT was carried out according to the Westminster Protocol without provocative pharmacological tests. In the first phase of the HUTT, the children were lying flat in a quiet and warm study room for 30 min. Then, after this time, the children were buckled to the tilt table and lifted to a vertical position (to an angle of 60°). The duration of verticalization was 45 min, after which, the child was returned to a flat position. Blood pressure measurements and heart rate evaluation were performed before verticalization, after verticalization, and at 5-min intervals throughout the verticalization process, and after the child was returned to a flat position. Head-up vasovagal responses were classified according to the new Classification by the Vasovagal Syncope International Study (VASIS classification): a mixed response was defined as hypotension followed by a decrease in HR that remained >40 b.p.m.; a cardioinhibitory response was defined as an HR decrease to <40 b.p.m. for >10 s and/or occurrence of asystole >3 s; and a vasodepressor response was defined as hypotension without an HR decrease of >10% from the peak HR prior to syncope [14].

### 2.3. Statistical Analysis

Data were collected in a database in a form of a Microsoft EXEL spreadsheet. The results were analyzed using the Minitab® program for statistical analysis (version 18.1, 1829 Pine Hal Rd, State College, PA 16801, USA). Discrete variables were expressed as proportions. Comparison between groups was made using the two proportion test. Evaluating results were considered if the tested sample data contained at least five observations (events) and the normal approximation *p*-value was a result. Otherwise, when the sample data contained less than five events, the Fisher exact method result was used. A *p*-value of <0.05 was considered statistically significant.

### 2.4. Ethical Approval

Approval from the Bioethical Commission of the Silesian Medical University in Katowice (KNW/0022/KB/116/15) was obtained prior to the commencement of this study.

## 3. Results

During the recruitment period (January 2013 and December 2016), we analyzed 825 medical records of children with diagnosed syncope. A total of 109 patients with final diagnosis VVS were included in this study. All these patients with recurrent syncope with including criteria were referred for HUTT evaluation, and 60 of these (55.1%) had a positive result of HUTT (+). A negative HUTT (−) result was observed in 49 (44.9%) children. The cause of admission to the cardiological yard is shown in Table 1. 

All patients underwent education regarding a change of lifestyle to prevent the recurrent syncopal episodes (*n* = 109/100%). Moreover, they were advised to take magnesium preparations (56/109 or 51.3%), remain under the cardiological (106/109 or 97.2%), neurological (65/109 or 59.6%), ophthalmological care (12/109 or 11%), and see a family physician (66/109 or 60.5%) as an outpatient. 

### 3.1. Patient Characteristics

The average age of the study group was 15.8 ± 1.5 years, and the majority were females (80/109 or 73.3%) The demographic characteristics of patients with VVS according to the HUTT response are reported in Table 2.

Sixty-eight of the study group with VVS were presyncope events (68/109 or 62.3%). The period of recurred syncopal during the 2–3 years was 39/109 (or 35.7%), ≤1 year in 41/109 (or 37.6%), and 29/109 (or 26.6%) in the last six months. Fainting occurred occasionally and several times a year in 62/109 (or 56.8%) of cases; once a year in 28/109 (or 25.6%); up to 1–2 times a month in 19/109 (or 17.4%); and only 5/109 or 4.5% had them most often, once or several times a week. The average of the hospitalization time was 5.2 ± 1.4 days. Twelve (12/109 or 11%) subjects experienced a head injury during a syncopal spell. The head injury for another cause than fainting was reported by six (6/109 or 5.5%) patients. A total of 13/109 or 11.9% children attended additional sporting activities (football, swimming, athletics, handball, judo, dance, volleyball). 

### 3.2. The Clinical Features Evaluation

We did not observe significant differences in the prodromal symptoms of syncope between both studied groups of children: VVS (+) and VVS (−). The clinical features of prodromal symptoms are shown in Table 3.

In a detailed history taking of accompanying symptoms of the syncope, cramps–contractures and speech disorders were statistically higher among VVS (−) than VVS (+) (*p* = 0.016, *p* = 0.038). However, only in the researched group with VVS (−), seizures (3/49 or 6.1%) with no difference statistically appeared. The additional symptoms reported only in VVS (+) subjects were body stiffness and urine incontinence (1/60 or 1.6%). 

The most important findings were the significant differences appeared in the predisposing factors of syncopal event. The fear–pain–stress emotion (16/49 or 32.6%; *p* = 0.02) was significantly greater in the VVS (−) group of children. We did not observe any statistical difference in circumstances of the occurrence of syncope between the VVS (+) and VVS (−) groups in the following factors: persistent standing (28.60 or 46.6% vs. 20.49 or 40.8%); the change in body position (33/60 or 55% vs. 13/49 or 26.5%); a sitting position (7/60 or 11.6% vs. 2/49 or 4%); a after effort (26/60 or 43.3% vs. 19/49 or 38.7%); and a bath (3/60 or 3.3% vs. 0/60 or 0%).

## 4. Discussion

Syncope in children and adolescents is still a challenging condition for practicing physicians. It is estimated that 25–50% of patients can be determined for the cause of syncope during the initial evaluation [15]. According to new guidelines for the diagnosis and management of syncope, a clinical nurse specialist plays an important role in structured history taking. The syncope unit clinical nurse specialist should have the necessary skills to deliver the assessment for syncope and should be skilled in the performance of structured history taking [1]. 

Using a detailed clinical history, in approximately in 60% of cases, syncope can be defined and differentiated from other forms of transient loss of consciousness [16]. Thus, the investigations of VVS features might be important in expanding the knowledge concerning syncopal syndromes, especially when it concerns confidence in syncope diagnosis such as vasovagal with negative HUTT (−). In the present study, HUTT accounted for more than half of the positive results (60/109 or 55.1%). However, the worth of this test has recently been questioned [1,17]. The HUTT may be helpful in distinguishing syncope from psychogenic syncope [18]. Moreover, HUTT is widely accepted as a useful tool to demonstrate the mechanism of vasovagal responses, especially a vasodepressive tendency [19]. Similarly, in this research, the most common type of vasovagal syncope was the vasodepressor reaction (31/109 or 51.7%). Vasodepressor syncope should be differentiated from orthostatic syncope. In young people, postural orthostatic tachycardia syndrome (POTS) can cause syncope, in which, apart from symptoms of brain perfusion during verticalization, it is accompanied by a significant acceleration of the heart rate by 30 b.p.m. (or >120/min) [20]. It happens that the perceived acceleration of heart rate is determined by patients as palpitations. In this study, heart palpitations occurred in both groups of children with VVS (+): 18/60 or 30% and VVS (−): 17/49 or 34.6% (*p* = 0.603). Furthermore, chest discomfort was observed in 18/60 or 30% in the VVS (+) group and in 9/49 or 18.3% in the VVS (−). Zhang et al. also found that chest discomfort occurred more often in children with VVS (17/55 or 30.9%) compared to children with cardiac syncope (3/31 or 9.7%) [21].

According to the new European Society of Cardiology (ESC) guidelines, post-exercise syncope is associated with vasovagal (reflex) syncope and syncope occurring during exercise may be of a cardiogenic effect [13]. In our study, the post-exercise syncope was not statistically different (*p* = 0.630), which shows that there is no difference in post-exercise symptom between the diagnostic profile of VVS with negative and positive HUTT because this factor is typical for vasovagal syncope.

Many potential diagnostic approaches of seizure and VVS remain difficult and may contribute to diagnostic errors. Epileptic seizures may be incorrectly diagnosed as syncope [1,22]. Seizures are consequential to deep cerebral ischemia, and secondary to hypotension and bradycardia. Typical episode of syncope could abruptly transform into prolonged clonic movements. Shorter epileptic seizures of which the duration is shorter than a few minutes may remain unnoticed [1,23]. In the present study, we observed syncopal symptoms in both study groups, mimicking and resembling neurological disorders such as numbness (3/60 or 3.3% vs. 2/49 or 4%), tinnitus (2/60 or 3.3% vs. 4/49 or 8.1%), and somnolence (1/60 or 1.6% vs. 1/49 or 2%). In 3/49 (or 6.1%) cases of VVS (−), seizures were reported. Speech disorders and cramps–contractures significantly occurred only in the group with VVS (−) (4/49 or 8.1%; *p* = 0.038, and 5/49 or 10.2%; *p* = 0.016). 

Zhang et al. found that urine incontinence in children with cardiac syncope was higher than that of those with vasovagal syncope (8/31 or 25.8% vs. 1/55 or 5.5%; *p* = 0.001) [21]. Similarly, in our study, we observed urine incontinence in one child with VVS (+) (1/60 or 1.6%)

According to Wieling et al., the prodromal signs and symptoms are more often experienced in young subjects when a spontaneous vasovagal syncope is imminent [2]. They found nausea and epigastric distress and sweating in typical premonitory symptoms for reflex syncope. These authors explained that these prodromes were related to more robust autonomic control. Additionally, Zhang et al. compared the characteristics of patients with VVS with those of cardiac syncope. They found that nausea–vomiting and sweating placed in pediatric patients with cardiac- and vasovagal syncope without a significant difference in both groups [21]. Similarly in our studies, nausea–vomiting and sweating occurred in both study groups without a statistically significant difference.

The lack of premonitory symptoms in about 30% patients who experience syncope creates difficulties in the initial clinical assessment [24]. In the research of McHarg et al, prodromal symptoms occurred in 68% of patients with syncope where dizziness accounted for the most frequent symptom [25]. Additionally, in our investigations, the most common prodromal symptom for both studied groups, VVS (+) and VVS (−), was dizziness (19/60 or 31.6%, and 19/49 or 38.7%). 

The initial evaluation based on history taking is important in diagnostic evaluation, management, and risk stratification. According to the ESC recommendations for the initial evaluation, the VVS is highly probable if syncope is caused by pain, fear, or standing, and is associated with typical prodromal symptoms such as pallor, sweating, and/or nausea. It is considered that patients with VVS have low-risk features, thus they are recommended to be discharged from the emergency department [1,26].

The major high-risk features in patients with syncope at the initial evaluation in the emergency department are new onset of chest discomfort, breathlessness, abdominal pain or headache, syncope during exertion or when supine, and sudden onset of palpitations [1]. In our study, all of these high-risk features were exposed without the abdominal pain. However, there were no other features that suggested a serious condition (past medical history, physical examination, other diagnostic tests). 

In the central type of syncope, a vasovagal reaction is triggered in the cerebral cortex and hypothalamus centers by stimulating sensory fibers in specific clinical situations like fear, blood phobia, pain, or strong emotions [1,6,27]. We found that the independent factor predicting the vasovagal syncope was the fear–pain stress emotion, which occurs in negative HUTT (−) (*p* = 0.016). Thus, the characteristic syncope factor that emotional stress belongs might confirm vasovagal syncope despite a negative HUTT result.

This was a retrospective study and patient data were obtained from archived medical files. The limitations of the present study include the small sample size of the analyzed group of children (*n* = 109). The study strictly evaluated the criteria when selecting patients who had been referred to the diagnostic and treatment facility for syncope. This research is the first step in expanding the knowledge in this area, thus in the future, we plan to collect more study patients in a multi-center study. The present study did not evaluate the specificity and sensitivity of HUTT in relation to those applied in other protocols with pharmacological provocation. Many potential diagnostic approaches to cover the clinical features of syncope depend on using methods of HUTT. O’Dwyer et al. determined the prevalence of amnesia for vasovagal syncope using the Italian Protocol of HUTT [28]. Therefore, we need to carry out further follow-up investigations to establish the difference between the result of the HUTT and the clinical features in relation to the type of protocol and larger study group. 

## 5. Conclusions

There is a close relation in the diagnostic profile between the negative and positive results of the head-up tilt table test in children with vasovagal syncope. Thus, collecting careful history taking from patients and eyewitnesses about the clinical features characterizing syncope play important roles in diagnosing VVS.

## Figures and Tables

**Table 1 ijerph-17-04524-t001:** Cases of syncope in children in the schedule of four years of hospital admissions.

Year	The Hospital Silesian Children’s Health Centre	Children’s Cardiology Unit	Children with Syncope Diagnosed (R-55) ^1^
2013	21,552	1720	192
2014	44,887	1675	207
2015	43,493	1758	210
2016	44,402	1832	216

^1^ World Health Organization International Statistical Classification of Diseases and Related Health Problems.

**Table 2 ijerph-17-04524-t002:** Comparison of the characteristics of children with vasovagal syncope (VVS) between the group with positive (+) and negative (−) head-up tilt table test (HUTT) results.

Characteristics	VVS (+) ^1^	VVS (−) ^2^
	(*n* = 60)	(*n* = 49)
Demographic		
Age	15.8 ± 1.5	15.8 ± 1.5
Male, *n* (%)	16 (26.7%)	13 (26.5%)
Female, *n* (%)	44 (73.3%)	36 (73.5%)
HUTT ^3^ response *n* (%)		
Mixed	27 (45%)	-
Vasodepressor	31 (51.7%)	-
Cardioinhibitory	2 (3.3%)	-

^1^ VVS (+): children with vasovagal syncope (VVS) and a positive result of head up tilt table test.^2^ VVS (−): children with vasovagal syncope and a negative result of head up tilt table test. ^3^ HUTT: head-up tilt table test.

**Table 3 ijerph-17-04524-t003:** Predictors of VVS in children and adolescents with positive and negative results of HUTT.

Clinical Feature	VVS (+) ^1^	VVS (−) ^2^	*p-*Value
	(*n* = 60)	(*n* = 49)	
Prodromal symptoms*, n* (%)			
Dizziness	19 (31.6%)	19 (38.7%)	0.439
Chest discomfort	18 (30%)	9 (18.3%)	0.151
Palpitations	18 (30%)	17 (34.6%)	0.603
Fatigue	17 (28.3%)	16 (32.6%)	0.626
Dyspnea	15 (25%)	15 (30.6%)	0.516
Blurred vision	14 (23.3%)	17 (34.6%)	0.193
Nausea, vomiting	3 (5%)	3 (6.1%)	1.000
Pale skin	2 (3.3%)	4 (8.1%)	0.405
Tinnitus	2 (3.3%)	4 (8.1%)	0.405
Body tremor	2 (3.3%)	3 (6.1%)	0.656
Sweating	1 (1.6%)	3 (6.1%)	0.324
Imbalance	1 (1.6%)	2 (4%)	0.587
Accompanying symptoms of syncope, *n* (%)			
Numbness	3 (3.3%)	2 (4%)	1.000
Body stiffness	1 (1.6%) *	-	1.000
Urine incontinence	1 (1.6%) *	-	1.000
Somnolence	1 (1.6%)	1 (2%)	1.000
Seizures	-	3 (6.1%)	0.088
Speech disorders	-	4 (8.1%)	0.038 *
Cramps-contractures	-	5 (10.2%)	0.016 *
Other symptoms, *n* (%)			
Headache	18 (30%)	15 (30.6%)	0.945
Reduced exercise tolerance	6 (10%)	10 (20.4%)	0.134
Predisposing factors*, n* (%)			
Persistent standing	28 (46.6%)	20 (40.8%)	0.539
Fear–pain stress emotion	8 (13.3%)	16 (32.6%)	0.016 *
Change of body position	33 (55%)	13 (26.5%)	0.292
Sitting position	7 (11.6%)	2 (4%)	0.182
After effort	26 (43.3%)	19 (38.7%)	0.630
Bath	3 (3%)	-	0.251

^1^ VVS (+): children with vasovagal syncope (VVS) and a positive result of head up tilt table test. ^2^ VVS (−): children with vasovagal syncope and a negative result of head up tilt table test. * Significant difference (*p* < 0.05) compared between the study groups, as calculated using the two proportion test.

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
