# Peer review of "The Differences in the Diagnostic Profile in Children with Vasovagal Syncope between the Result of Head-Up Tilt Table Test"

_ijerph, 2020, doi:10.3390/ijerph17124524_

Round 1

Reviewer 1 Report

The study of Kolarczyk et al. is disigned and performed well but the manuscript needs extencive editing of English language and style.

1.  The sentence in lines 49-51 should be rearranged properly.

2.  Line 76: "a initial" should be "an initial".

3.  The sentence in lines 81-82 should be rearranged properly.

4.  Line 83: "To participate of" should be "To participate in".

5.  Line 90: "and" should be present before "exclusion".

6.  Line 102: "," should be added after "seconds".

7:  Line 103: "with-out" should be "without".

8.  Line 120: "admission on" should be "admission to".

9.  The paragraph in lines 124-127 should be rearranged properly.

10. Line 138: "Sixty-eight of study" should be "Sixty eight of the study".

11. Line 143: "Twele" should be "Twelve".

12. The sentence in line 144 should be rearranged properly.

13. Line 149: "group" should be "groups".

14. The paragraph in lines 155-159 should be rearranged prperly.

15. Line 161: "occurs" should be deleted.

16. Line 162: "," should be added after "group", and "in" after "was" should be deleted.

17. Line 164: "of" should be added after "circumstances", and "the" should be added after "in".

18. The paragraph in lines168-173 should be rarranged properly.

19. The two sentences in lines 174-177 should be rearranged properly.

20. Lines 177-178: "in present" should be "in the present".

21. Line 179: "has" should be added before "recently", and "distinguish" should be "distinguishing".

22. Page 6 is full with English language, grammatical, and style errors almost in every line. It should be edited and rearranged properly by an expert in English language.

23. The paragraph of the conclusions section should be edited and rearranged properly by an expert in English language.

24. The references should be rearranged in accordence with the style of the Int J Environ Res Public Health.  

Author Response

Dear Reviewer,

We appreciate the insightful review and interesting inquiries.

We agree with your suggestions  that the manuscript needs extensive language and style editing. The English style was improved.

Dear Reviewer

Once again, we want to thank you for the review and we hope that our answers are satisfying.

Best regards,

Ewelina Kolarczyk

Reviewer 2 Report

Kolarczyk et al. reported the difference in clinical characterizing in children with VVS.  I can not understand the results and conclusions.  The first question is about analysis. the sample size of your study is small. If the analysis results were no significant, can you describe " no difference in statistically"? Second, how we can use the results of your study in clinical practice?  Can we select patients before the HUTT? The third question is about prodromal symptoms. In your dates, typical symptoms with VVS such as Nausea and sweating are few.  Please describe the reasons for the discussion. The last is about the diagnosis.  What did you diagnose the patients in VVS(-) ?  Are all patients in VVS(+) diagnosed VVS?
Please improve your Manuscript.   

Author Response

Dear Reviewer,

We appreciate the insightful review and interesting inquiries, very much.

Your four questions (major points):

  1. The first question is about analysis. the sample size of your study is small. If the analysis results were no significant, can you describe " no difference in statistically"?

Our answer:

We will change these sentences and describe it according to Reviewer’s suggestion to:  " no difference in statistically".

Our study group was isolated in a process of careful selection and checked against the compatibility of the criteria.  Out of 825 medical hospitalization records over the period of 4 years time, we chose the total of (notice Table 1 page 3, line 122) 109 children diagnosed with VVS who were included in the study group. We are aware of the small sample size of our study group, thus we included it in limitation in the discussion section (page 6, line 206-207 in the first draft for peer review). This research is the first step to expanding the knowledge in this area, thus in the future we  are planning to collect more study patients as a multi-center study.

  1. Second, how we can use the results of your study in clinical practice?  Can we select patients before the HUTT?”

Our answer:

The selection of patients before the HUTT in Emergency Department is recommended by the European Society of Cardiology (ESC guidelines 2018) – notice page 2, line 49-51; 54-57 and 60-65 in the first manuscript for peer review. The knowledge of prodromal symptoms and the circumstances in patients with syncope in history taking plays an important role in initial assessment in the Emergency Department. The  initial evaluation helps to:

  1. establish whether there was indeed a syncope,
  2. the physicians can differentiate syncope from other forms of transient loss of consciousness (TLOC)
  3. appropriate management in the Emergency Department
  4. usefulness in risk stratification
  5. appropriate selection to avoid the unnecessary high costs of specialized diagnostic tests
  6. according to the new ESC guidelines (2018), the initial evaluation in VVS is more important than tilt test – the HUTT should be considered only a means of exposing a hypotensive tendency rather than being diagnostic of VVS.
  7. The third question is about prodromal symptoms. In your dates, typical symptoms with VVS such as Nausea and sweating are few.  Please describe the reasons for the discussion.”

Our answer:

Nausea and sweating are referred to on page  6, line 208-212 in the first manuscript for peer review. But we can improve that:

According to Wieling et al the prodromal sings and symptoms are more often experienced in  young subjects when a spontaneous vasovagal syncope is imminent. They found nausea and epigastric distress and sweating in typical premonitory symptoms for reflex syncope. These authors explain that these prodromes are related to more robust autonomic control (Heart 2004,90:1094-1100).

Also Zhang et al, compared the characteristics of patients with VVS with cardiac syncope.  They found that nausea- vomiting and sweating placed in pediatric patients with cardiac – and vasovagal syncope without  a significant difference in both groups (Cardiology in Young 2013, 23:54-60).

Similarly in our studies, nausea-vomiting and sweating occurred in both study groups without a statistically significant difference.

4.” The last is about the diagnosis.  What did you diagnose the patients in VVS(-) ?  Are all patients in VVS(+) diagnosed VVS?

We administered analysis of medical records of children and adolescents who were diagnosed  with vasovagal syncope. It is extensively explained in the materials and methods section (page 2 line 67-90 at first draft for peer review).

All studied patients in VVS (+) were diagnosed VVS, and all studied patients with VVS (-) were diagnosed with vasovagal syncope.

Negative HUTT (-) does not exclude vasovagal syncope diagnosis. The main indication of its use is the confirmation of the diagnosis of reflex syncope, because it allows induction of vasovagal reaction and  shows the mechanism of this reaction (vasodepressor, cardioinhibitory and mixed). It is also a useful tool to distinguish vasovagal syncope from other kinds of TLOC, and from psychogenic pseudosyncope. According to the latest ESC recommendations, the usefulness of HUTT has declined in importance in the diagnosis of vasovagal syncope. The value of history taking of prodromal symptoms in identifying children with vasovagal syncope plays a key role in the evaluations of syncope (ESC 2018).   In our study we explained it in “Discussion” section (page 5, line 172-181 at first draft for peer review).

Dear Reviewer

Once again, we would like to thank you for the review and we hope that our answers are satisfactory.

Ewelina Kolarczyk

Reviewer 3 Report

The authors presented that the differences of the clinical profile in children with vasovagal syncope between the HUT positive and HUT negative results. The result of the study is interesting and may be useful in the clinical practice. However, there is some concern for acceptance.

Major points:

  1. The authors mentioned that the most important findings were the different rates of predisposing factors including “the fear-pain-stress emotion” and “after effort syncope”. However, the numbers and rates of “the fear-pain-stress emotion” in HUT positive group and that of “after effort syncope” in HUT negative group were not appeared in the manuscript and tables. The authors need to show the predisposing factors and circumstances of syncopal events in the new table. In addition, “after effort syncope” is not appeared in abstract.
  2. Discussion is too long and redundant. The authors should focus the points of discussion including predisposing factors such as “the fear-pain-stress emotion” and “after effort syncope”, and accompanying symptoms. The authors need to comment why emotional syncope was prevalent in HUT negative group and post-effort or post-exercise syncope was prevalent in HUT positive group in discussion.
  3. Introduction is also too long and redundant. Please reduce the volume of the description.
  4. Conclusions are not attractive. The authors need to reconsider the conclusions presented one or two sentences briefly.
  5. As the authors mentioned, the numbers of study population are very limited. The authors are expected to collect more study patients as a multi-center study.

Minor points:

There are several mis-spelled in the manuscript. Be careful for spelling.

Page 4, line 143: mis-spelled “Twelve” as “Twele”

Page 5, line 155: mis-spelled “accompanying” as “accopanying”

Page 5, line 158: mis-spelled “reported” as “reportet”

Page 5, line 160: mis-spelled “predisposing” as “predisporing”

Page 5, line 162: mis-spelled “greater” as “grater”

Author Response

Dear Reviewer,

We appreciate the insightful review and interesting inquiries very much.

Your five questions (major points):

  1. The authors mentioned that the most important findings were the different rates of predisposing factors including “the fear-pain-stress emotion” and “after effort syncope”. However, the numbers and rates of “the fear-pain-stress emotion” in HUT positive group and that of “after effort syncope” in HUT negative group were not appeared in the manuscript and tables. The authors need to show the predisposing factors and circumstances of syncopal events in the new table. In addition, “after effort syncope” is not appeared in abstract”.

Our answer:

The numbers and the rates of “the fear-pain-stress emotion” and “after effort syncope” in both study groups: VVS (+) and VVS(-) are described in the text of page 5 line 159-161.  According to the guidelines for authors, we didn’t want to repeat these things twice, so the table is ommited.

“the fear-pain-stress emotion” and “after effort syncope” are  the circumstances of syncope that are conducive to occur syncope - not the prodromal symptoms.

Of course, according to the Rewiever’s suggestions, we will add the circumstances of syncope in a Table and in the abstract.

  1. Discussion is too long and redundant. The authors should focus the points of discussion including predisposing factors such as “the fear-pain-stress emotion” and “after effort syncope”, and accompanying symptoms. The authors need to comment why emotional syncope was prevalent in HUT negative group and post-effort or post-exercise syncope was prevalent in HUT positive group in discussion”.

Our answer:

(in the first draft for peer review).

The comment about “the fear-pain-stress emotion” is included in the discussion: page 6, line 219-224.

The accompanying symptoms are described in discussion, notice:

Page 5:

heart palpitations -line 185,

chest discomfort - line 187-189.

Page 6:  numbness – line 196;

tinnitus – line197;

somnolence – line 197,

speech disorders – line 198

cramps-contractures – line 198

dizziness – line 205

pain, fear, standing, pallor, sweating, nausea – line 209-211.

According to the ESC guidelines, “post-effort” (or “post-exercise syncope”) is associated with vasovagal (reflex) syncope and syncope occurring during exercise may be a cardiogenic effect (Polish Heart Journal 2009, 67(12): page 570).  In our study, the “post-effort” or “post-exercise syncope” was not statistically significant: VVS (+) 26 or 43.3%; VVS (-) 19 or 38.7%; p=0.630, which shows that there is no difference in post-exercise symptom between diagnostic profile of VVS with negative and positive HUTT, because this factor is typical for vasovagal syncope.

According to the Rewiever’s suggestions, we will add the comment about the post-effort syncope but It is very difficult to reduce the discussion - especially as the Reviewer himself suggests adding elements to the discussion.

3.” Introduction is also too long and redundant. Please reduce the volume of the description”.

Our answer:

According to the Rewiever’s suggestions, we will reduce the introduction.

  1. Conclusions are not attractive. The authors need to reconsider the conclusions presented one or two sentences briefly”.

Our answer:

According to the Reviewer’s suggestions, we changed the conclusions to the following:

“There is a close relation in the diagnostic profile between the negative and positive result of Head-up Tilt table test in children with vasovagal syncope”

  1. “As the authors mentioned, the numbers of study population are very limited. The authors are expected to collect more study patients as a multi-center study”.

Our answer:

Our study group was isolated in a process of careful selection and checked against the compatibility of the criteria.  Out of 825 medical hospitalization records, over the period of 4 years time we chose the total of 109 children diagnosed with VVS who were included in the study group.  We are aware of the small sample size of our study group, thus we included it in limitation in the discussion section (page 6, line 206-207 in the first draft for peer review). This research is the first step to expanding the knowledge in this area, thus in the future we  are planning to collect more study patients as a multi-center study.

Minor points:

“There are several miss-spelled in the manuscript. Be careful for spelling.”

Our answer:

Thank you for your suggestion. We improved the English language and style.

Dear Reviewer

Once again, thank you for the review and we hope that our answers are satisfactory.

Ewelina Kolarczyk

Round 2

Reviewer 2 Report

The manuscript was revised adequately.

Author Response

Dear Reviewer,

We are very appreciate for the insightful comment.

We definitely agree with You and we improved our manuscript according Your remark. Our improved version is below:

“The most important findings were the significant differences appeared in the predisposing factors of syncopal event. The fear-pain-stress emotion (16/49 or 32.6%; p=0.02) was significantly greater in VVS (-) group of children. We did not observe no difference in statistically in circumstances of the occurrence of syncope between VVS (+) and VVS (-) groups in the following factors: persistent standing (28.60 or 46.6% vs. 20.49 or 40.8%), the change of a body position (33/60 or 55% vs. 13/49 or 26.5%), a sitting position (7/60 or 11.6% vs. 2/49 or 4%), a after effort (26/60 or 43.3% vs. 19/49 or 38.7%) and a bath (3/60 or 3.3% vs. 0/60 or 0%)”.

We also removed the sentence describing syncope after exercise from the abstract.

Our improved in the main document are marked in green color.

Dear Reviewer

Once again, thank You for the review and we hope that our answers are satisfactory.

Ewelina Kolarczyk

Reviewer 3 Report

The authors have corrected the revised manuscript and answered the reviewer’s questions properly. The revised version seems to be acceptable except one concern mentioned later. The authors need to correct this issue for acceptance.

Page 5, line 151-152: the sentence “after effort syncope was significantly higher (p=0.02) in VVS (+) group of children.” is not correct. Table 3 shows that after effort syncope was not different (p=0.63) between VVS (+) and VVS (-) groups. Therefore, in abstract, the sentence describing “post-exercise syncope” (page 1, lime 30-31) can omit.

Author Response

Dear Reviewer,

We are very appreciate for the insightful comment.

We definitely agree with You and we improved our manuscript according Your remark. Our improved version is below:

“The most important findings were the significant differences appeared in the predisposing factors of syncopal event. The fear-pain-stress emotion (16/49 or 32.6%; p=0.02) was significantly greater in VVS (-) group of children. We did not observe no difference in statistically in circumstances of the occurrence of syncope between VVS (+) and VVS (-) groups in the following factors: persistent standing (28.60 or 46.6% vs. 20.49 or 40.8%), the change of a body position (33/60 or 55% vs. 13/49 or 26.5%), a sitting position (7/60 or 11.6% vs. 2/49 or 4%), a after effort (26/60 or 43.3% vs. 19/49 or 38.7%) and a bath (3/60 or 3.3% vs. 0/60 or 0%)”.

We also removed the sentence describing syncope after exercise from the abstract.

Our improved in the main document are marked in green color.

Dear Reviewer

Once again, thank you for the review and we hope that our answers are satisfactory.

Ewelina Kolarczyk
